# A Novel Dynamic Light-Section 3D Reconstruction Method for Wide-Range Sensing

**DOI:** 10.3390/s24123793

**Published:** 2024-06-11

**Authors:** Mengjuan Chen, Qing Li, Kohei Shimasaki, Shaopeng Hu, Qingyi Gu, Idaku Ishii

**Affiliations:** 1School of Advanced Science and Technology, Hiroshima University, Higashihiroshima 739-8527, Japan; chenmengjuan2016@ia.ac.cn (M.C.); soleilor@mail.tsinghua.edu.cn (Q.L.); shimasaki@robotics.hiroshima-u.ac.jp (K.S.); hu@robotics.hiroshima-u.ac.jp (S.H.); 2Institute of Automation, Chinese Academy of Sciences, Beijing 100190, China; qingyi.gu@ia.ac.cn

**Keywords:** dynamic 3D reconstruction, multi galvanometers, light section, calibration

## Abstract

Existing galvanometer-based laser-scanning systems are challenging to apply in multi-scale 3D reconstruction because of the difficulty in achieving a balance between a high reconstruction accuracy and a wide reconstruction range. This paper presents a novel method that synchronizes laser scanning by switching the field-of-view (FOV) of a camera using multi-galvanometers. Beyond the advanced hardware setup, we establish a comprehensive geometric model of the system by modeling dynamic camera, dynamic laser, and their combined interaction. Furthermore, since existing calibration methods mainly focus on either dynamic lasers or dynamic cameras and have certain limitations, we propose a novel high-precision and flexible calibration method by constructing an error model and minimizing the objective function. The performance of the proposed method was evaluated by scanning standard components. The results show that the proposed 3D reconstruction system achieves an accuracy of 0.3 mm when the measurement range is extended to 1100 mm × 1300 mm × 650 mm. This demonstrates that for meter-scale reconstruction ranges, a sub-millimeter measurement accuracy is achieved, indicating that the proposed method realizes multi-scale 3D reconstruction and simultaneously allows for high-precision and wide-range 3D reconstruction in industrial applications.

## 1. Introduction

Light-section vision systems are widely used in many applications for their adaptability, high accuracy, and effective cost [1,2,3], such as rail traffic monitoring [4], medical imaging [5], robotics [6], and industrial production [7]. Such systems typically comprise a camera, laser projector, and mechanical scanning platform. The line laser projects laser stripes onto the surface of the object, whereas the camera captures an image of the object with the laser stripes. The three-dimensional (3D) geometric information of the object is then obtained by triangulation, as extensively reviewed in the literature [8]. The 3D reconstruction of an object can be completed by passing laser stripes or objects through a mechanical scanning platform.

Traditional laser scanners rely primarily on mechanical driver shafts, which are large, complex, and slow [9,10]. To overcome these limitations, various scanning mechanisms have been proposed. For instance, Du [2] designed a system that mounts a line laser on the end of a robotic arm to improve scanning flexibility; however, its scanning accuracy is limited by the precision of the robotic arm. Jiang [11] proposed a system that uses gimbals to drive the laser and camera for scanning; however, the system’s size is significant, and its scanning speed is slow. In recent years, galvanometers have emerged as promising scanning devices because of their small size, fast rotation, and high control accuracy. This galvanometer-based solution provides a better alternative in terms of laser-scanning accuracy and speed [12]. However, existing galvanometer-based laser-scanning systems are primarily designed to perform laser scanning while leaving the camera fixed. The limited FOV of a fixed camera causes a trade-off between the accuracy and the sensing range of the system, which significantly affects its efficiency.

In this study, we propose a novel dynamic light-section 3D reconstruction system that combines a dynamic laser and dynamic camera using multi-galvanometers. Our approach utilizes multiple galvanometers to synchronize laser scanning and the FOV switching of the camera, thereby enabling high-precision and wide-range 3D reconstruction. Calibration is required to achieve this, which includes the system calibration of the galvanometer-based dynamic laser and camera, and their joint calibration. For calibrating galvanometer-based dynamic laser systems, Eisert [13] introduced a geometric model and calibration procedure; however, the model is complicated, and its optimization is difficult, thus leading to a low accuracy. Yu [14] designed a one-mirror galvanometer laser scanner. However, the calibration procedure is complex, and the objective function is difficult to optimize. Similarly, Yang [15] proposed a calibration method based on a precision linear stage. However, this approach relies on a precision instrument and lacks flexibility. For calibrating galvanometer-based dynamic camera systems, Ying et al. [16,17,18] introduced self-calibration methods, which are complex in theory and difficult to implement. Kumar [19] proposed a calibration method based on the look-up table (LUT) using simple linear parameters, which requires complex pre-processing. Junejo et al. [16,17,20,21] proposed feature-based calibration methods, which are time-consuming and have a low accuracy. Han [22] introduced a calibration method for galvanometer-based camera using an end-to-end single-hidden-layer feed forward neural network model, but it is computationally intensive. Boi et al. [23,24] proposed manifold constrained Gaussian process regression methods for galvanometer setup calibration, which rely on data-driven and complex calibration procedures. Hu [25] built a galvanometer mirror-based stereo vision measurement system and established a mirror reflection model, but it still lacks an accurate calibration method.

In conclusion, current light-section 3D reconstruction systems cannot simultaneously have a high accuracy and wide range. Moreover, existing calibration methods only focus on calibrating either dynamic lasers or dynamic cameras and still have some shortcomings, as mentioned above. To address these limitations, this study proposes a novel dynamic 3D reconstruction system that overcomes the trade-off between accuracy and measurement range by synchronizing laser scanning and the FOV switching of a camera based on multiple galvanometers. Additionally, we propose a novel comprehensive calibration solution for the proposed system, encompassing the calibration of the dynamic camera, dynamic laser, and their joint calibration. The contributions of this study can be summarized as follows:(1)A novel dynamic light-section 3D reconstruction system is designed based on multiple galvanometers. To the best of our knowledge, this system is the first to synchronize laser scanning and the FOV switching of camera, thus enabling high-precision and wide-range 3D reconstruction simultaneously.(2)A novel high-precision and flexible calibration method for the dynamic 3D system is proposed by constructing an error model and objective function based on the combined model of the dynamic camera and dynamic laser. This method is not only applicable to the proposed system but also to other single galvanometer-based dynamic laser or camera systems.(3)Experiments were conducted to validate the proposed dynamic 3D reconstruction method and demonstrate its accuracy. To the best of our knowledge, compared to all existing galvanometer-based laser-scanning methods, our approach has the highest measurement range while maintaining the same level of measurement accuracy.

The system design and geometric model are described in Section 2. The proposed calibration method and error compensation methods are described in Section 3. Section 4 presents the validation experiments and results. Finally, Section 5 presents the conclusions.

## 2. System Design and Geometric Model

### 2.1. System Design

The dynamic light-section 3D reconstruction system consists of a CMOS camera, a line laser, and two galvanometer mirror systems, as shown in Figure 1. The camera and Galvanometer-1 form a dynamic camera system, whereas the laser and Galvanometer-2 form a dynamic laser system.

Based on the geometric model of the system and pre-calibration, the 3D information of the target can be calculated from the captured laser image and voltage values of the two galvanometers. The working principle is illustrated in Figure 2. A spherical object on a flat plane is employed to demonstrate the process of dynamic 3D reconstruction. First, the system utilizes multi-galvanometer control to scan the target surface. When the system is activated, a line laser projects a laser stripe onto Galvanometer-2, which reflects the stripe onto the surface of the object. By controlling the voltage of Galvanometer-2, the laser stripe can scan the target. Simultaneously, the dynamic camera system captures laser images from different angles by adjusting the voltage of Galvanometer-1. Next, the laser-center-pixel coordinates are obtained using the laser stripe extraction algorithm. The 3D reconstruction of the laser stripe is performed by combining the voltage values of multiple galvanometers, laser pixel coordinates, geometric models of the dynamic 3D reconstruction system, and calibrated parameters. The point clouds of all the laser stripes are converted to the same coordinate frame using the transformation matrix of the dynamic camera. The system performs error correction based on the joint calibration to optimize accuracy. Finally, the system generates a point cloud for the target and completes the dynamic 3D reconstruction. Accurate geometric modeling and calibration methods are essential to ensure the 3D reconstruction accuracy of the system.

### 2.2. Geometric Model

The geometric model of the system is shown in Figure 2b. When the system is precisely machined, we can assume that for the dynamic camera system, the optical axis of the camera is perpendicular to the rotation axis of Galvanometer-1’s pan mirror and incident at the center of the pan mirror. For the dynamic laser system, the laser plane is perpendicular to the rotation axis of Galvanometer-2’s pan mirror and incident on the center of the mirror. Four coordinate frames are established. {W} is the world coordinate frame defined on the surface of the planar chessboard target, with the origin point O located at the upper-left corner of the chessboard. The X-axis and Y-axis are parallel to the chessboard array, while the Z-axis is perpendicular to the O-XY plane, following the right-hand coordinate system. {C} represents the camera coordinate frame, where the Z-axis corresponds to the camera’s optical axis, and the X-axis and Y-axis are parallel to the image plane, following the right-hand coordinate frame. {G} denotes the coordinate frame of Galvanometers-1, with the Z-axis corresponding to the rotation axis of the pan mirror. The X-axis is aligned with the camera’s optical axis, while the Y-axis aligns with the line connecting the center points of the pan and tilt mirrors. $\left\{V_{t}\right\}$ is the virtual camera coordinate frame formed by the reflection of {C} through the pan and tilt mirrors of Galvanometers-1. According to the operating principle of a galvanometer, the rotation angle of the pan–tilt mirror is proportional to the voltage. For Galvanometer-1, the voltages of the pan–tilt mirrors are denoted by $U_{1-pan}$ and $U_{1-tilt}$. Therefore, the rotation angle of the pan mirror is $\theta_{1}=k_{1-pan}U_{1-pan}$, and the rotation angle of the tilt mirror is $\theta_{2}=k_{1-tilt}U_{1-tilt}$. As the pan–tilt mirror rotates, $\left\{V_{t}\right\}$ reflects the change in $U_{1-pan}$ and $U_{1-tilt}$. When $U_{1-pan}=U_{1-tilt}=0$, the initail virtual camera coordinate frame is denoted by $\left\{V_{0}\right\}$. The relationship between $\left\{V_{t}\right\}$ and $\left\{V_{0}\right\}$ is given by Equation (Equation 1).
(1)TV0Vt=TGVtTV0G,
where TV0Vt is the transformation matrix between $\left\{V_{0}\right\}$ and $\left\{V_{t}\right\}$. TGVt denotes the transformation matrix between {G} and $\left\{V_{t}\right\}$. TV0G denotes the transformation matrix between $\left\{V_{0}\right\}$ and {G}. As shown in the geometric model diagram, {C} is first reflected by a pan mirror and then by a tilt mirror. The geometries of the two reflections are modeled using Equation (Equation 3). For $\left\{V_{0}\right\}$, the rotation angle of the pan–tilt mirror is $\theta_{1}=\theta_{2}=45^{\circ }$. The transformation matrix TV0Vt is calculated using Equation (Equation 2). Thus, the geometric model of the dynamic camera is established.
(2)TV0Vt=TGVtTG−1V0|(U1−pan=U1−tilt=0).
(3)TGVt=10000cos2θ2sin2θ2d(1−cos2θ2)0sin2θ2−cos2θ2−dsin2θ20001−cos2θ1sin2θ100sin2θ1cos2θ10000100001001−l100001000001=sin2θ10−cos2θ1lcos2θ1cos2θ1cos2θ2sin2θ2sin2θ1sin2θ2−lsin2θ1cos2θ2+d(1−cos2θ2)cos2θ1sin2θ2−cos2θ2sin2θ1sin2θ2−lsin2θ1sin2θ2−dsin2θ20001.

The coordinates of the pixel points on the laser stripe are denoted by (u,v). The coordinates of the corresponding 3D points in $\left\{V_{t}\right\}$ are denoted by $\left(X_{V},Y_{V},Z_{V}\right)$. Based on the pinhole model of the camera, the mapping relationship between their coordinates can be obtained using Equation (Equation 4).
(4)$$Z_{V}\left[\begin{array}{c} u\\ v\\ 1 \end{array}\right]=\left[\begin{array}{ccc} f_{x} & 0 & u_{0}\\ 0 & f_{y} & v_{0}\\ 0 & 0 & 1 \end{array}\right]\left[\begin{array}{c} X_{V}\\ Y_{V}\\ Z_{V} \end{array}\right],$$
where $\left(u_{0},v_{0}\right)$ is the principal point of the image, and $f_{x}$ and $f_{y}$ are the focal length of the camera. For Galvanometer-2, the rotation angle of the pan mirror is denoted by $\theta_{3}=k_{2-pan}U_{2-pan}$, and the rotation angle of the tilt mirror is denoted by $\theta_{4}=k_{2-tilt}U_{2-tilt}$. When $U_{2-pan}=U_{2-tilt}=0$, the dynamic laser is in its initial position. The initial laser plane in $\left\{V_{0}\right\}$ is denoted by plane0V0, and its equation is $A_{0}x+B_{0}y+C_{0}z+D_{0}=0$. The rotation axis of the dynamic laser in $\left\{V_{0}\right\}$ is denoted as $\vec{n}=\left(n_{x},n_{y},n_{z}\right)$. The plane of the laser after rotation about the rotation axis is denoted as plane0V0, and the equation is $A_{V_{0}}x+B_{V_{0}}y+C_{V_{0}}z+D_{V_{0}}=0$.

The light path of the dynamic laser is reflected by a mirror and rotated along its axis. The rotation angle of the laser plane is twice that of the mirror plane. Therefore, the equation for the dynamic laser plane after rotation in $\left\{V_{0}\right\}$ can be solved using Equation (Equation 5).
(5)$$\left[\begin{array}{c} A_{V_{0}}\\ B_{V_{0}}\\ C_{V_{0}} \end{array}\right]=R\left(\vec{n},2\theta_{4}\right)\left[\begin{array}{c} A_{0}\\ B_{0}\\ C_{0} \end{array}\right],$$
where R represents the Rodrigues transformation. Using a point $P=(X_{0},Y_{0},Z_{0})$ on the rotation axis, $D_{V_{0}}$ can be calculated as $D_{V0}=-A_{V_{0}}X_{0}-B_{V_{0}}Y_{0}-C_{V_{0}}Z_{0}$. Combining this with the transformation matrix TV0Vt in Equation (Equation 2), the equation for the dynamic laser plane in $\left\{V_{t}\right\}$ can be calculated as
(6)planeVt=TV0VtplaneV0.

The equation for planeVt is denoted as $A_{V}x+B_{V}y+C_{V}z+D_{V}=0$. By extracting the pixel points (u,v) from the laser stripe, the corresponding 3D point can be calculated as PVt=(XV,YV,ZV). Therefore, the relationship between PVt and the change in the galvanometer mirror angles can be expressed by Equation (Equation 7).
(7)$$\left\{\begin{array}{c} Z_{V}=D_{V}/\left[A_{V}\times \left(u-u_{0}\right)/f_{x}+B_{V}\times \left(v-v_{0}\right)/f_{y}+C_{V}\right]\\ X_{V}=\left(u-u_{0}\right)/f_{x}\times Z_{V}\\ Y_{V}=\left(v-v_{0}\right)/f_{y}\times Z_{V}\\ \left\{A_{V},B_{V},C_{V},D_{V}\right\}=F\left(\theta_{1},\theta_{2},\theta_{4};A_{0},B_{0},C_{0},D_{0},l,d,\vec{n},P\right). \end{array}\right.$$

F represents a parameterized mapping function, where $\theta_{1},\theta_{2},\theta_{4}$ are variables and other parameters are constants. Because $\left\{V_{t}\right\}$ changes constantly with the scanning angle, it is necessary to convert all PVt into a coordinate frame that is fixed with {W}. For ease of calculation, we choose $\left\{V_{0}\right\}$ and obtain PV0=(XV0,YV0,ZV0)=TVtV0PVt.

Thus, we establish the relationship between (u,v) and $\left(X_{V_{0}},Y_{V_{0}},Z_{V_{0}}\right)$ to formulate the 3D reconstruction. In the geometric model of the 3D dynamic system, Equation (Equation 7) shows that $f_{x},f_{y},u_{0},v_{0}$ can be obtained by calibrating the camera. $A_{0},B_{0},C_{0},D_{0}$ can be obtained from the laser plane calibration. $l,d,\vec{n}=\left(n_{x},n_{y},n_{z}\right),P=\left(X_{0},Y_{0},Z_{0}\right)$ are unknown, and a calibration algorithm must be designed to obtain the unknowns.

## 3. Calibration Method

The proposed system calibration method is divided into three parts: the dynamic camera calibration, the dynamic laser calibration, and the joint calibration of the dynamic camera and laser for error correction.

### 3.1. Dynamic Camera Calibration

Firstly, we perform intrinsic parameter calibration using Zhang’s method [26] for distortion correction and obtain the camera parameters $f_{x},f_{y},u_{0},v_{0}$. The dynamic camera geometric model described in Section 2 is used to determine the constraint relationship between $\left\{V_{t}\right\}$ and {G}, as shown in Equation (Equation 3); thus, we can obtain the parameters *l* and *d*. The proposed dynamic camera calibration method uses a large calibration board as shown in Figure 2b. The calibration board measures 740 × 740 mm and comprises a total of 35 × 35 circular markers. These markers are constructed from 25 individual 7 × 7 sub-patterns. Each sub-pattern features a central larger circular marker with a diameter of 15 mm, while the remaining smaller circular markers have a diameter of 10 mm, with a center-to-center spacing of 20 mm. The purpose of the larger circular markers is to establish the relationships between calibration points across different FOVs. The calibration board is scanned by varying the galvanometer voltage to obtain numerous images at different rotation angles. Each image corresponds to a virtual coordinate frame. The number of images is denoted as *n*. The transformation matrix between $\left\{V_{t}\right\}$ and {W} is obtained through Zhang’s extrinsic parameter calibration method [26], denoted as TWV0,TWV1,TWV2,…,TWVn. As the relative positions between the calibration points in these images are known, the transformation matrix between the virtual coordinate frames is calculated as TV0V1,TV0V2,…,TV0Vn. These values are taken as observations. Multiple sets of observations are used to solve for the parameters to be calibrated. The initial pan–tilt angles of Galvanometer-1 are denoted by $\theta_{1}^{\left(0\right)}$ and $\theta_{2}^{\left(0\right)}$. They are the corresponding angles of $\left\{V_{0}\right\}$ and the first calibration image. The pan–tilt angles of Galvanometer-1 corresponding to $\left\{V_{n}\right\}$ and *n*th calibration images are denoted as $\theta_{1}^{\left(n\right)}$ and $\theta_{2}^{\left(n\right)}$. Therefore, $\left\{V_{0}\right\}$ and $\left\{V_{n}\right\}$ are defined as follows:(8)TGV0=TGVt|θ1=θ1(0),θ2=θ2(0)=kij(θ1(0),θ2(0),l,d),TGVn=TGVt|θ1=θ1(n),θ2=θ2(n)=gij(θ1(n),θ2(n),l,d),
where $k_{ij}$ and $g_{ij}$ represent the function of TGV0 and TGVn. *l* and *d* are parameters to be calibrated using Equation (Equation 7). The transformation matrix between $\left\{V_{0}\right\}$ and $\left\{V_{n}\right\}$ is a 4 × 4 matrix, which can be expressed as follows:(9)TV0Vn(n)=a11na12na13na14na21na22na23na24na31na32na33na34n0001=aijn.

Simultaneously, TGVn can be calculated using TGV0 from Equation (Equation 8) and TV0Vn. For ease of representation, this is denoted as $h_{ij}$.
(10)TGVn=TV0Vn(n)·TGV0=∑i=14∑j=14aijnkijθ1(0),θ2(0),l,d=hijaijn,θ1(0),θ2(0),l,d

Equation (Equation 10) is the result of $\left\{V_{t}\right\}$ obtained from multiple observations. Equation (Equation 8) shows the results calculated using the geometric model of the dynamic camera. For all the measured coordinate frames ($\left\{V_{0}\right\},\left\{V_{1}\right\},\left\{V_{2}\right\},\ldots ,\left\{V_{n}\right\}$), the sum of the errors between the theoretical and measured values must be minimized. Therefore, the objective function is formulated using Equation (Equation 11).
(11)$$\begin{array}{cc} \; & l^{*},d^{*}=\underset{l,d}{\arg\min}\sum_{t=1}^{n}\sum_{i=1}^{4}\sum_{j=1}^{4}\\ \; & {\left[h_{ij}\left(a_{ij}^{n},\theta_{1}^{\left(0\right)},\theta_{2}^{\left(0\right)},l,d\right)-g_{ij}\left(\theta_{1}^{\left(n\right)},\theta_{2}^{\left(n\right)},l,d\right)\right]}^{2}. \end{array}$$

In Equation (Equation 3), the parameters *l* and *d* exist only in a translation vector. Based on the objective function, Equation (Equation 12) can be obtained. Finally, the parameters *l* and *d* are obtained by solving Equation (Equation 12) using the least-square method.
(12)$$\left\{\begin{array}{c} g_{14}\left(\theta_{1}^{\left(1\right)},\theta_{2}^{\left(1\right)},l,d\right)-h_{14}\left(a_{ij}^{1},\theta_{1}^{\left(0\right)},\theta_{2}^{\left(0\right)},l,d\right)=0\\ g_{24}\left(\theta_{1}^{\left(1\right)},\theta_{2}^{\left(1\right)},l,d\right)-h_{24}\left(a_{ij}^{1},\theta_{1}^{\left(0\right)},\theta_{2}^{\left(0\right)},l,d\right)=0\\ g_{34}\left(\theta_{1}^{\left(1\right)},\theta_{2}^{\left(1\right)},l,d\right)-h_{34}\left(a_{ij}^{1},\theta_{1}^{\left(0\right)},\theta_{2}^{\left(0\right)},l,d\right)=0\\ \ldots \\ g_{14}\left(\theta_{1}^{\left(n\right)},\theta_{2}^{\left(n\right)},l,d\right)-h_{14}\left(a_{ij}^{n},\theta_{1}^{\left(0\right)},\theta_{2}^{\left(0\right)},l,d\right)=0\\ g_{24}\left(\theta_{1}^{\left(n\right)},\theta_{2}^{\left(n\right)},l,d\right)-h_{24}\left(a_{ij}^{n},\theta_{1}^{\left(0\right)},\theta_{2}^{\left(0\right)},l,d\right)=0\\ g_{34}\left(\theta_{1}^{\left(n\right)},\theta_{2}^{\left(n\right)},l,d\right)-h_{34}\left(a_{ij}^{n},\theta_{1}^{\left(0\right)},\theta_{2}^{\left(0\right)},l,d\right)=0. \end{array}\right.$$

### 3.2. Dynamic Laser Calibration

Based on the geometric modeling of the dynamic laser presented in Section 2, the equation of plane0V0 and the rotation axis of the dynamic laser $\vec{n}$ must be calibrated when Galvanometer-1 is at the initial position. The calibration of plane0V0 is conducted using the methodology outlined in [27]. This process involves the acquisition of laser images at various positions using a checkerboard calibration plate. Through this approach, the laser plane is accurately defined by fitting multiple laser lines. For the extraction of the laser center, the technique described in [28] is employed. This algorithm, based on the Hessian matrix, can achieve sub-pixel precision extraction and a processing speed of 1350 frames per second. Following these procedures, we successfully derive the equation $A_{0}x+B_{0}y+C_{0}z+D_{0}=0$.

The Galvanometer-2 voltages $U_{2-tilt}=U_{1},U_{2},\ldots ,U_{m}$ are used to move the laser and obtain multiple laser planes. Next, the respective equations are calibrated in the same manner as plane0V0 and denoted as plane1V0,plane2V0,…,planemV0. The unit normal vectors of these planes are calculated as ${\vec{n}}_{0}\left(n_{x_{0}},n_{y_{0}},n_{z_{0}}\right)$, ${\vec{n}}_{1}\left(n_{x_{1}},n_{y_{1}},n_{z_{1}}\right)$, ${\vec{n}}_{2}\left(n_{x_{2}},n_{y_{2}},n_{z_{2}}\right),\ldots ,$${\vec{n}}_{m}\left(n_{x_{m}},n_{y_{m}},n_{z_{m}}\right)$. In the absence of errors, the laser planes intersect along the same straight line. This line is the laser rotation axis $\vec{n}$ and is also the rotation axis of the tilt mirror in Galvanometer-2. For a normal vector in any laser plane, we obtain $\vec{n}\cdot {\vec{n}}_{i}=0$(i=0,1,2,…,m). However, $\vec{n}\cdot {\vec{n}}_{i}$ is not exactly equal to zero because of various errors. Therefore, for all laser planes, the objective function is formulated as shown in Equation (Equation 13).
(13)$$\begin{array}{c} {\vec{n}}^{*}=\underset{n_{x},n_{y},n_{z}}{\arg\min}\left(\sum_{i=0}^{m}{\left(n_{x}n_{x_{i}}+n_{y}n_{y_{i}}+n_{z}n_{z_{i}}\right)}^{2}\right). \end{array}$$

The direction vector $\vec{n}$ of the rotation axis is obtained by minimizing the objective function. $P=(X_{0},Y_{0},Z_{0})$ is a point on the rotation axis in all laser planes and can be obtained using the least-square method. All parameters in Equation (Equation 7) are obtained by calibrating the camera, laser plane, dynamic camera, and dynamic laser. Thus, we complete the calibration of the proposed dynamic 3D system.

### 3.3. Joint Calibration for Error Correction

For a well-calibrated dynamic light-section 3D reconstruction system, there are two sources of error, dynamic camera and dynamic laser, as listed in Table 1.

This study proposes an error-correction method based on the joint calibration of a dynamic camera and dynamic laser. After the calibration is completed, error correction is performed based on the 3D reconstructed results. Theoretically, when Galvanometer-1 is scanning and Galvanometer-2 is fixed, the reconstructed laser point cloud coincides perfectly. However, as explained in the error source analysis, there is some deviation between the multiple laser point clouds owing to these errors. We correct these errors using point-cloud registration based on the Iterative Closest Point (ICP) algorithm [29] to obtain the accurate transformation matrix.

The calibration process is designed based on the following principle. If Galvanometer-2 remains stationary, the laser will be out of the FOV after Galvanometer-1 has scanned a certain range. Therefore, multiple calibration positions must be set in advance to maintain the laser in the FOV. These are set in advance as $p_{1},p_{2},\ldots ,p_{n}$, and the corresponding voltages of Galvanometer-2 at these positions are $\left(s_{1},t_{1}\right),\left(s_{2},t_{2}\right),\ldots ,\left(s_{n},t_{n}\right)$, respectively. The laser plane equations $plane_{p_{1}},plane_{p_{2}},\ldots ,plane_{p_{n}}$ at these positions are calibrated using the laser plane calibration method described in Part B. The voltages of Galvanometer-1 are denoted by ($s_{1}^{\prime }$, $t_{1}^{\prime }$), ($s_{2}^{\prime }$, $t_{2}^{\prime }$),…,($s_{m}^{\prime }$, $t_{m}^{\prime }$), respectively. The error-correction flow is presented in Algorithm 1.
**Algorithm 1** Joint Calibration for Error Correction  1:**Input:** $f_{x},f_{y},u_{0},v_{0}$; l,d; $A_{1},B_{1},C_{1},D_{1},\ldots ,A_{n},B_{n},C_{n},$$D_{n}$; $\left(s_{1},t_{1}\right),\left(s_{2},t_{2}\right),\ldots ,\left(s_{n},t_{n}\right)$.  2:**Output:** TV1V0,TV2V0,…,TVmV0.  3:**Initialize:** *i*← 1, (s,t)←$\left(s_{1},t_{1}\right)$, $\left(s^{\prime },t^{\prime }\right)$←f(s,t), du ← 0.1, planeV1←$plane_{p_{1}}$, M← Unit matrix.  4:Using Equation (Equation 7) to obtain the laser point cloud P1V0.  5:**while** i<m **do**  6:     Laser Image capture and center curve extraction;  7:     planeVt=g(s,t,s′,t′);  8:     Using Equations (1)–(3) to calculate TVtV;  9:     planeV←TVtVplaneVt;10:     Using Equation (Equation 7) to obtain the point cloud PiV0.11:     TVtV← Transformation matrix between P1V0 and PiV0;12:     TVtV0←$M^{V}T_{V_{t}}$;13:     **if** i%50==0 **then**14:         (s,t)←$\left(s_{\left(i/50\right)},t_{\left(i/50\right)}\right)$, planeVt←Vplane;15:         Using Step. (8)–Step. (11) to obtain point cloud P.16:         M← Transformation matrix between P and PiV0;17:     **end if**18:     i←i+1;19:     $\left(s^{\prime },t^{\prime }\right)$$\leftarrow (s^{\prime }+du\times i\%200,t^{\prime }+1\times \;int\left(i/200\right))$;20:**end while**

## 4. Experiment

The proposed dynamic 3D reconstruction system was built as shown in Figure 3. The camera model was MV-CA004-10UC, with a pixel size of 6.9 µm × 6.9 µm, resolution of 720 pixels × 540 pixels, and frame rate of 500 fps. The exposure time of the camera to capture the dark image of the laser was 500 µs. The laser model was LXL65050-16, and the laser wavelength was 650 nm. The models of both Galvanometer-1 and Galvanometer-2 were TSH8310. The galvanometer was used to scan a range of $\pm 20^{\circ }$ using a control voltage range from −10 V to +10 V. The maximum scan frequency was 1 kHz, with an angular resolution of 0.0008; thus, the system had the potential for a high accuracy and resolution.

### 4.1. Calibration Accuracy Verification

#### 4.1.1. Dynamic Camera Calibration Accuracy

Twenty-five images of the calibration board were collected when $U_{1-pan}=U_{1-tilt}=0$. The camera was calibrated using Zhang’s [26] calibration method in OpenCV. After calibration, the intrinsic parameters were $f_{x}=$ 7801.38, $f_{y}=$ 7798.24, $u_{0}=$ 359.51, and $v_{0}=$ 269.54. The focal length was 53.83 mm. According to the calibration method for the dynamic cameras presented in Section 3, the system parameters were solved as *l* = 83.45 mm and *d* = 22.14 mm. Based on these calibration results, the geometric model proposed in Section 2 can be used to calculate the theoretical transformation matrix for the pan–tilt mirror of Galvanometer-1 at different angles.

The transfer matrices corresponding to these angles are directly measured using a calibration board. The matrix 2-norm is calculated according to Equation (Equation 14) to compare the theoretical matrices A and measured transfer matrices B for the calibration accuracy verification. The pan–tilt voltages of Galvanometer-1 are varied from −10 V to +10 V at intervals of 4 V. Thirty-six positions are measured. The error between the theoretical and measured transfer matrices is obtained, and the error curves are shown in Figure 4a. The results show that the RMSE (Root Mean Square Error) is 1.231 mm between the theoretical and measured values.
(14)$$E\left(A,B\right)=\sqrt{\sum_{i=1}^{4}\sum_{j=1}^{4}{\left(A_{ij}-B_{ij}\right)}^{2}}$$

This confirms the accuracy of the dynamic camera calibration. The observed errors originate from the geometric model and the calibration process, as explained in the error analysis section. It is important to note that the measured values obtained for the virtual camera using the calibration board may also exhibit slight deviations. Consequently, these findings serve as a validation of the accuracy of the dynamic camera calibration; however, these cannot be solely relied upon to assess the accuracy of the calibration. A more detailed accuracy verification can be conducted based on the outcomes of the 3D reconstruction analysis.

#### 4.1.2. Dynamic Laser Calibration Accuracy

With Galvanometer-1 fixed, the calibration board was positioned within the FOV of the virtual camera. The tilt mirror of Galvanometer-2 was rotated 30 times with a step size of 0.1 V, allowing the system to scan the calibration board, whose position was randomly changed 5 times (ensuring clear imaging in the virtual camera); the same 30 scans were repeated for each position. The laser rotation axis is solved as $\left(\vec{n},P\right)=\left(\left[0.99,0.02,-0.0004\right],\left[-18310.30,-195.93,257.97\right]\right)$. Figure 4b visually represents the laser plane and the rotation axis. Notably, the calibrated rotation axis align with the intersection of the laser planes, providing evidence for the accuracy of the dynamic laser calibration. A detailed accuracy assessment is subsequently performed by analyzing the results of the 3D reconstruction.

#### 4.1.3. Joint Calibration Accuracy

A calibration sphere was selected as the 3D reconstruction target for error correction. Galvanometer-2 was controlled to project the laser stripe onto the sphere, while Galvanometer-1 was fixed. The virtual camera, controlled by Galvanometer-1, captured images of the laser stripe from different views. The 3D reconstruction of these laser stripe images was performed based on the calibration results and geometric models of the 3D dynamic system. The reconstructed point clouds, which are indicated as white, are shown in Figure 2a, ’Error Correction’. Notably, white point clouds exhibit non-overlapping regions owing to errors. The correction method described in Algorithm 1 is employed to register these white point clouds. The registration results are shown as colored point clouds in Figure 2a. The distance between the point clouds before and after the correction is calculated to evaluate the error. The calculation formula is as follows:(15)$$d=\frac{1}{|P_{s}|}\sum_{i=1}^{\left|P_{s}\right|}{∥p_{t}^{i}-p_{s}^{i}∥}^{2},$$

Here, $p_{s}$ represents the point cloud of the laser stripe captured in the first virtual camera view, and $p_{t}$ represents the point cloud of the laser stripe captured from another view. $|P_{s}|$ denotes the number of points in $p_{s}$. The error between $p_{t}$ and $p_{s}$ is determined by performing a nearest-neighbor search, denoted as Error1. After the point-cloud registration, the error between $p_{t}$ and $p_{s}$ is calculated as Error2. In addition, the error before correction is computed as Error3 using the matched points from the point-cloud registration result. The error curves are shown in Figure 4c. The RMSE of Error1 and Error3 before correction are calculated as 4.928 and 5.475 mm, respectively. However, after error correction, the RMSE of Error2 is significantly reduced to 0.197 mm. These results evidently indicate a substantial improvement in the accuracy following the error-correction process.

### 4.2. 3D Reconstruction Accuracy Verification

#### 4.2.1. Standard Blocks Reconstruction Test

A standard stair block was employed to test the stability of the system and analyze its reconstruction accuracy at different angles. The stair block had a distance of 30 mm between its two planes, with machining errors within 1 µm. The 3D dynamic system proposed in the paper was used to reconstruct a stair block. Scanning was performed by synchronously controlling the tilt mirrors of both Galvanometer-1 and Galvanometer-2, rotating each by 0.1°. Once the scanning and reconstruction processes were completed, a point cloud of the stair block was generated. Two planes (Plane-1 and Plane-2) of the stairs were fitted, and the distance between them was calculated. Point clouds belonging to Plane-2 were used to fit a plane equation using the least-square method. Next, 500 points belonging to Plane-1 were randomly selected, and the average distance between these points and Plane-2 was calculated as the distance of the fitted plane. The difference between the calculated and actual distances is considered as the error, which serves as a measure of the reconstruction accuracy achieved by the system.

The reconstruction distance is 650 mm. The FOV for a single virtual camera is 120 mm × 120 mm, while the dynamic camera’s FOV expands to 1300 mm × 1300 mm (including a 10% overlap area for better stitching), thus enlarging the camera’s imaging range by a factor of 117.4. The scanning range of the dynamic laser is 1100 mm × 1640 mm. The measurement range of the system is determined by the overlapping FOV of the dynamic camera and dynamic laser, which measures 1100 mm × 1300 mm. Thirty different positions are selected to analyze the reconstruction accuracy at different angles. The dynamic camera and laser simultaneously scan the target from these positions to complete the 3D reconstruction process. Figure 5a shows examples of reconstructions obtained from four different positions, providing a visual representation of the reconstructed 3D models. The thickness error, which is related to the rotation angles of Galvanometer-1 and Galvanometer-2, is analyzed, as shown in Figure 5b. It is evident from the graph that the error in 3D reconstruction increases as the rotation angles of the galvanometers deviate from their initial positions (calibration position). This is because the camera’s focal length is adapted to the calibration position, and imaging areas far from the calibration position may become blurred due to defocusing, thereby affecting accuracy. Furthermore, as analyzed in Table 1, errors due to various reasons accumulate more as the distance from the calibration position increases. The RMSE for these thirty positions is calculated as 0.165 mm. These values provide evidence of the high precision achieved by the proposed system for 3D reconstruction.

#### 4.2.2. Accuracy and Reconstruction Range Comparison with Existing Methods

To compare the performance of the proposed method with that of existing methods [12,14,15,30,31,32,33], we conducted comparative experiments using the standard component scanning method. The accuracy of the dynamic light-section 3D system depends primarily on the working distance. To perform a fair comparison, we repeated the standard component scanning procedure at various reconstruction distances, namely, 100, 200, 350, 400, and 1000 mm, which are consistent with the working distances employed in existing methods. As analyzed in the previous experiment, the system’s reconstruction accuracy is related to the scanning position. In order to reduce errors other than calibration errors, we placed the target at the center of the scanning area.

The 3D reconstruction accuracy, the measurement ranges of the traditional methods, and the magnification factor of the proposed method’s reconstruction range compared to traditional methods are presented in Table 2. From the obtained results, it can be concluded that the proposed method exhibits smaller errors and larger measurement ranges than the existing methods at the corresponding working distances. This demonstrates the superior performance of our method in terms of accuracy and range compared with existing methods.

#### 4.2.3. Large Object Scanning Test

The large high-precision machined flat plate and sphere were also used to test the 3D reconstruction accuracy. The size of the plate was 740 mm × 740 mm, and the sphere had a diameter of 350 mm. Similar to Experiment B (1), the system scanned the target and obtained its point clouds with the scanning distance set at 650 mm, and the 3D measurement range was 1100 mm × 1300 mm. The position and angle of the target were arbitrarily changed within the scanning range, and the reconstruction was repeated three times for each target. For the plate target’s point cloud, a plane equation was fitted using the RANSAC algorithm. The distances between all the points and the fitted plane were calculated, and the average of these distances was considered the error of the dynamic 3D system reconstruction. For the sphere target’s point cloud, a sphere was fitted based on the RANSAC algorithm, and the diameter was calculated. The difference between the calculated diameter and the sphere’s actual diameter was taken as the error of the dynamic 3D reconstruction. The RMSE values for the three measurements were 0.281 mm for the plate and 0.226 mm for the sphere, respectively.

This experimental result demonstrates that the system achieves sub-millimeter reconstruction accuracy within a meter-scale reconstruction range, indicating that the system enables high-precision multi-scale 3D reconstruction across a wide-range reconstruction area. Moreover, a limitation of this system is that the dynamic camera extends the FOV using a galvanometer. When the extension angle is large, there can be some defocus, resulting in blurred images and subsequently affecting the accuracy of the 3D reconstruction. To address this issue, image deblurring algorithms can be used to improve imaging quality, thereby enhancing the reconstruction accuracy at the edges of the FOV.

## 5. Conclusions

A dynamic light-section 3D reconstruction system is proposed in this paper, which overcomes the trade-off between accuracy and measurement range by using multiple galvanometers. A geometric model of the system was established, and a flexible and accurate calibration method was developed. The experimental results demonstrate that the proposed system performs well in terms of measurements, indicating its potential for industrial applications where high-precision and wide-range 3D reconstruction is required. Furthermore, the proposed method can be used in conjunction with an active tracking system for 3D reconstruction of moving targets. This will be introduced in our subsequent work.

## Figures and Tables

**Figure 1 sensors-24-03793-f001:**
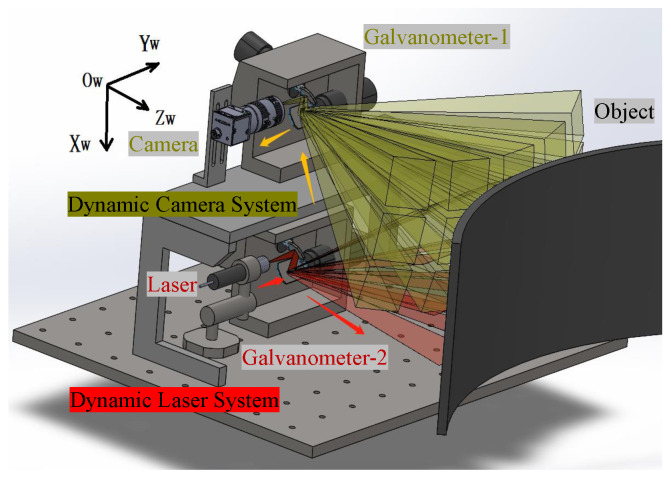
System design.

**Figure 2 sensors-24-03793-f002:**
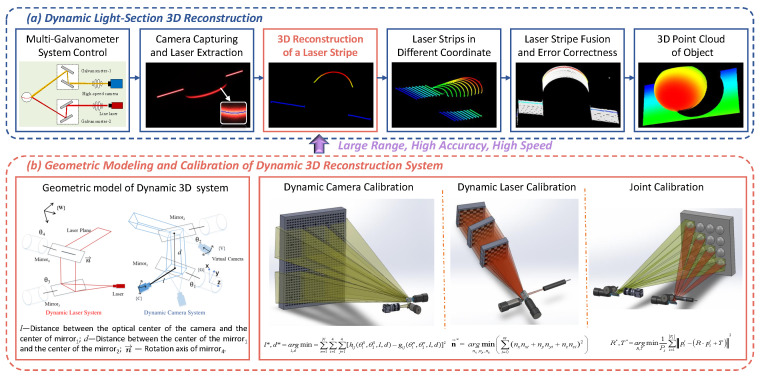
Framework of dynamic 3D reconstruction method. (**a**) The flowchart of dynamic light-section 3D reconstruction. (**b**) Geometric modeling and calibration of dynamic 3D reconstruction system.

**Figure 3 sensors-24-03793-f003:**
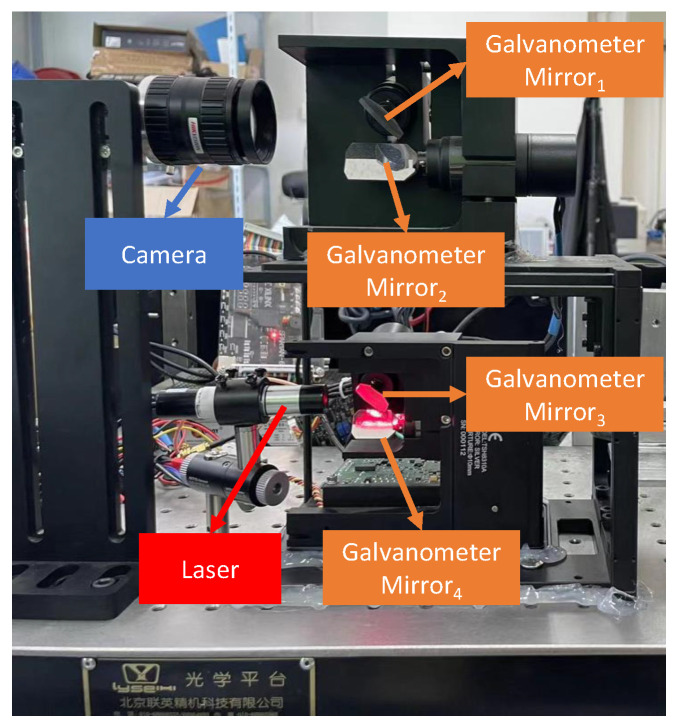
The 3D dynamic reconstruction system based on multiple galvanometers and light section.

**Figure 4 sensors-24-03793-f004:**
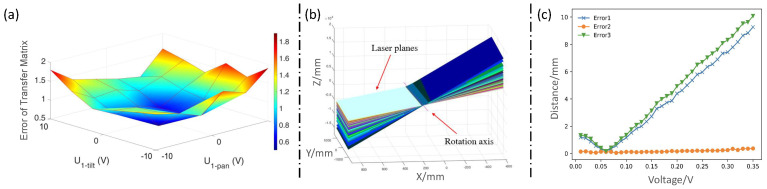
Calibration accuracy verification. (**a**) Error of dynamic camera transformation matrix. (**b**) Visualization of laser planes and the calibrated rotation axis. (**c**) Error curve before and after correction.

**Figure 5 sensors-24-03793-f005:**
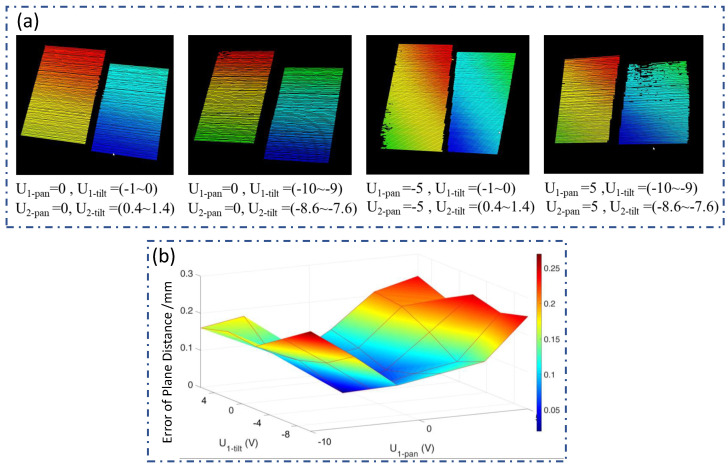
Standard blocks reconstruction test at different angles. (**a**) The point clouds of the stair at different angles. (**b**) 3D reconstruction error distribution at different angles.

**Table 1 sensors-24-03793-t001:** Error Source and Analysis.

	Error	Source and Analysis
1	Rotation angle of galvanometer mirror (θ1,θ2)	Non-linear deviations exist between voltage and rotation angle of Galvanometer-1.
2	Dynamic camera geometric model	The spectacular reflection geometric model has deviations with mechanical structure.
3	Calibration of parameter (l,d)	This error has been optimized by the proposed error model and objective function in this paper.
4	Camera intrinsic parameters calibration	This non-linear error is optimized using Zhang’s [26] calibration method.
5	Rotation angle of galvanometer mirror (θ3,θ4)	Non-linear deviations exist between voltage and rotation angle of Galvanometer-2.
6	Dynamic laser geometric model	This error depends on the accuracy of the laser’s mechanical installation.
7	Calibration of laser rotation axis	This error is optimized by the proposed objective function.
8	Laser center curve extraction	Center extraction algorithm is based on the Hessian Matrix and ensures a high extraction accuracy.

**Table 2 sensors-24-03793-t002:** Accuracy and reconstruction range comparison of traditional and proposed methods.

Working Distance/mm	Traditional Methods			Proposed Method		
	Name Year	Accuracy/mm	Range/mm	Accuracy/mm	Range/mm	Factor of Expanded Range
100	NOM-LSS 2017 [16]	0.01	10 × 10	0.01	130 × 200	260
250	3DM-LS 2018 [17]	0.1	80 × 80			
	EAC-LSS 2020 [14]	0.061	80 × 80	0.057	350 × 500	27.3
350	IS-LSS 2020 [30]	0.08	200 × 200	0.08	500 × 700	8.75
400	FFV-LSS 2007 [32]	0.222	150 × 200			
	FLR-LSS 2022 [31]	0.1	150 × 200	0.092	650 × 800	17.3
1000	U3D-LSS 2016 [33]	0.382	200 × 300	0.314	1600 × 2000	53.3

## Data Availability

Data are contained within the article.

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
