# Peer review of "A Novel Dynamic Light-Section 3D Reconstruction Method for Wide-Range Sensing"

_sensors, 2024, doi:10.3390/s24123793_

Round 1

Reviewer 1 Report

Comments and Suggestions for Authors

- "Is theta 3 equal to TG? I recommend putting it directly into formula 3."

- p. 181: Does the camera have no lens distortion? If not, it's necessary to mention it; if it does, explain why it hasn't been implemented

- A more precise description of how the laser trace is extracted from the image is missing.

- In Table 2, please add the resolution at which the listed systems are capable of scanning.

- I assume that all images/tables will be as readable in print as they are now only in digital form.

- Question for the authors: What are the main advantages of the listed scanners compared to traditional triangulation scanners based on the principle of structured light? These are inherently simpler, more accurate, faster, and can handle a larger volume. What is the cost of the tested system or systems in Table 2? This information is missing in the introduction.

I kindly ask the authors to answer my questions.

Reviewer 2 Report

Comments and Suggestions for Authors

The authors proposed a method that combines a dynamic camera system consisting of a galvanometer and a camera with a dynamic laser system consisting of a galvanometer and a line laser. A dynamic contour 3D reconstruction system was established, enabling the 3D reconstruction of large-scale targets with fairly high accuracy. The proposed method is of theoretical meaning and practical values.

There are still some issues in the manuscript need to be fixed,including:

1.     The characters and formulas need careful check and corrections. For instance 1) A subscript “i" is omitted in Eq.(2); 2) The usage of “n” in Eq.(11) is inappropriate;  and 3) It seems that the subscripts 'i' in Vi and the 'i' in kij, gij​ express different meanings in Equation (8), which is confusing.

2.     A more strong analysis is needed to explain the reason why the method in Section 3.3 can effectively eliminate or reduce the impact of the error sources summarized in Table 1, especially the error sources indexed 1,2,5,and 6.

3.     In Section 2.2, what are the respective meanings of the geometric model and the mathematical model? The first sentence of Section 2.2 states that the mathematical model is shown in Figure 2(b), but Figure 2(b) is labeled as the geometric model.

4.     The coordinate axes lack units in Figure 4(b).

5.     Are there any limitations of the proposed method? It is suggested to give a discussion.

Comments on the Quality of English Language

Moderate editing of English language is required.
